# Simultaneous control of carrier transport and film polarization of emission layers aimed at high-performance OLEDs

Masaki Tanaka [1] ✉, Chin-Yiu Chan [2], Hajime Nakanotani [2,3] & Chihaya Adachi [2,3] ✉

The orientation of a permanent dipole moment during vacuum deposition results in the occurrence of spontaneous orientation polarization (SOP). Previous studies reported that the presence of SOP in organic light-emitting diodes (OLEDs) lowers electroluminescence efficiency because electrically generated excitons are seriously quenched by SOP-induced accumulated charges. Thus, the SOP in a host:guest-based emission layer (EML) should be finely controlled. In this study, we demonstrate the positive effect of dipole-dipole interactions between polar host and polar emitter molecules on the OLED performance. We found that a small-sized polar host molecule that possesses both high molecular diffusivities and moderate permanent dipole moment, well cancels out the polarization formed by the SOP of the emitter molecules in the EML without a disturbance of the emitter molecules' intrinsic orientation, leading to high-performance of OLEDs. Our molecular design strategy will allow emitter molecules to pull out the full potential of the EMLs in OLEDs.

In recent years, thermally activated delayed fluorescence (TADF) molecules have been recognized as a promising emitter in OLEDs[1,2]. In fact, nearly 100% of internal EL quantum efficiency has been achieved in various TADF-OLEDs by harvesting both electrically generated singlet and triplet excitons for electroluminescence (EL). However, their operational stability has lagged behind the well-established conventional fluorescence-based OLEDs[3,4]. This is primarily attributed to the high-density accumulation of long-lived triplet excitons in TADF emitters during OLED operation. The accumulated $T_1$ excitons cause unwanted triplet-triplet and/or triplet-polaron annihilations that successively generate high-energy hot triplet excitons and polarons, resulting in the degradation of the organic molecules[5–7]. Since organic host-guest systems with an emitter molecule as a diluted guest have been widely used in OLEDs primarily due to their high photoluminescence quantum yield (PLQY), the selection of host molecules is

also crucial to improving the OLED performance. Here, the host molecules not only suppress the molecular aggregation of guest molecules but also control the carrier balance in emission layers (EMLs)[4]. Therefore, precise host engineering is crucial for realizing ultimate OLEDs with high EL efficiency and long operational stability.

Exciton relaxation and charge carrier transport/injection properties in OLEDs are critically affected by not only individual intrinsic molecular properties but also the aggregated film properties. Especially, the importance of film polarization has been pointed out by several recent advanced studies. When an organic molecule possesses permanent dipole momentum (PDM), there is a possibility to form spontaneous orientation polarization (SOP) in the vacuum-deposited films, in which a giant surface potential (GSP) is built along with the normal to the substrate surfaces[8,9]. Further, the SOP in films generates the corresponding polarization charges on top- and bottom-side

[1]Department of Biotechnology and Life Science, Tokyo University of Agriculture and Technology, 2-24-16 Naka-cho, Koganei, Tokyo 184-8588, Japan. [2]Center for Organic Photonics and Electronics Research (OPERA), Kyushu University, 744 Motooka, Nishi-ku, Fukuoka 819-0395, Japan. [3]International Institute for Carbon Neutral Energy Research (WPI-I2CNER), Kyushu University, 744 Motooka, Nishi-ku, Fukuoka 819-0395, Japan. ✉e-mail: m-tanaka@me.tuat.ac.jp; adachi@cstf.kyushu-u.ac.jp

interfaces of the films. Such interfacial charges can facilitate carrier injection from electrodes, and the charge accumulation at the interface between the organic layers having different SOP happens even at the voltage below the turn-on voltage[10–12]. However, the accumulated charges at the interfaces can seriously quench the electrically-generated excitons[13] and the internal electric field induced by the SOP dissociates the excitons[14], hence lowering the external EL quantum efficiency (EQE) in OLEDs. The OLED operational stability is also strongly influenced by the SOP because the accumulated charges at the organic interfaces cause exciton-polaron quenching (EPQ) to generate high-energy excited polarons acting as degradation sources. Thus, a smaller polarization is beneficial to enhance the device stability to avoid the degradation resulting from the EPQ[15,16]. However, the device stability relates not only to the film polarization but also to several factors, such as carrier transport and recombination properties, and these influence the density distributions of excitons and polarons in EMLs. To achieve ultimately high device stability, simultaneous device designs of SOP, carrier transport, and recombination properties are essential.

The formation mechanism of SOP has not been fully clarified yet, and the material selection is rather limited[17,18]. Also, the research on the GSP effect on OLED performance has just started. Interestingly, some preceding reports disclosed that higher substrate temperature decreases the GSP of deposited films such as Alq$_3$ and TPBi, and vice versa[13,19], while controlling substrate temperature affects not only GSP but also film density[20,21]. Thus, the impact of the GSP effect on OLEDs is rather complicated. Another proposed method to cancel out the SOP of an electron transport layer was diluting polar molecules with medium-density polyethylene (MDPE). The MDPE-codeposition reduced the SOP of a TPBi-ETL and improved the OLED stability due to the suppressed EPQ in a fluorescent EML[15]. This method would be beneficial to cancel out the SOP of organic layers. However, it may be difficult to apply to EMLs because the EML designs in conventional OLEDs are based on the precise controls of exciton and carrier performance such as fluorescence/phosphorescence yields, exciton confinement, carrier recombination, and carrier transport. Further, in a host-guest codeposited system as conventional EMLs, the molecular orientations of both host and guest molecules intricately influence the GSP of the deposited film, e.g., doped nonpolar molecules into polar matrices suppress dipole-dipole interactions between the polar molecules, affecting the net polarization of the codeposited films[22]. Therefore, to control the GSP of EMLs, the effect of both host and dopant emitter molecules should be investigated separately.

In this study, we explored methodology to control the SOP of an EML containing a polar guest molecule exhibiting TADF and a carbazole (Cz)-based host. The surface potential of the EMLs was investigated by a Kelvin probe method. Not only the GSP value but also the polarity strongly depends on the host matrix while the neat TADF film shows positive GSP. We uncovered that the most small-sized polar host molecule well cancels out the GSP that is built by guest molecules' orientation. Further, the carrier transport properties of the nonpolar EML were simultaneously improved due to suitable energy level differences and molecular packings between host and guest molecules. Hence, OLEDs consisting of the nonpolar EML exhibited the highest operational stability in the tested TADF-OLEDs. Additionally, stable blue-emitting TADF-assisted fluorescence OLEDs were achieved by tuning EML properties using the cohost technique. Our findings provide a clear host-selection strategy to control EML polarization and develop OLEDs with ultimate performance.

## Results and discussion

The molecular structure of a guest TADF molecule, HDT-1, is shown in Fig. 1a. We have already reported a high EQE of over 25% and stable device operation in the OLED with an HDT-1 (guest):3,3′-di(9H-carbazol-9-yl)−1,1′-biphenyl (mCBP) (host) based EML[23]. In this study, we compared OLED performance using the series of Cz-based host matrices namely 1,3,5-tris(3,6-diphenylcarbazole-9-yl)benzene (3DPCz), 1,3-bis(3,6-diphenylcarbazole-9-yl) benzene (2DPCz), and 3,6-diphenylcarzole-9-ylbenzene (1DPCz)[24]. The PLQY of a 10 mol% HDT-1 doped in a 1DPCz film was 90% which is comparable to that of an mCBP host film (PLQY = 95%), while the PLQYs of doped films with 2DPCz and 3DPCz were 82% and 61%, respectively. Since the T$_1$ energy of 1DPCz (2.80 eV) in the neat film is higher than that of HDT-1 (2.70 eV, Supplementary Fig. 1), the 1DPCz matrix can confine HDT-1 triplet excitons, resulted in the high PLQY. Indeed, the mCBP-based and the 1DPCz-based films shared the comparable PL decay profiles (Fig. 1b). The reduced delayed PL components of the 2DPCz and 3DPCz-based films indicate that the exciton confinement is not sufficient in these hosts due to the lower T$_1$ levels[24]. The films with 1DPCz and 2DPCz showed very slightly redshifted PL spectra (Fig. 1c), which can be attributed to the stronger polarization of the host molecules[25].

Carrier transport properties of the codeposited films were investigated using hole-only devices (HODs) and electron-only devices (EODs). Figure 2a and b show the current density (J) -voltage (V) characteristics of the HODs and EODs (the device structures and the energy level diagrams of the highest occupied molecular orbital (HOMO) and the lowest unoccupied molecular orbital (LUMO) are shown in Supplementary Fig. 2 and Fig. 2c, respectively). The mCBP and the 1DPCz-based EODs showed a higher electron current than those of other host molecules. On the other hand, 1DPCz- and 2DPCz-based HODs exhibited better hole transport properties than those of other host molecules. Since the hole transport of the neat HDT-1 film was low compared to the codeposited films with 1DPCz and 2DPCz, the better hole transport properties can be attributed to the HOMO level differences between the host and the emitter molecules. The HOMO levels of 1DPCz and 2DPCz are shallower than that of HDT-1 (Fig. 2c), indicating that the hole transport can be governed by the hole transport of 1DPCz and 2DPCz with the shallower HOMO level, leading to the improved hole current. In contrast, the HOMO levels of mCBP and 3DPCz were comparable to that of HDT-1, thus, the J-V curves of the HODs with these host molecules and the neat TADF were also comparable. On the other hand, the electron transports of the host:guest systems were considerably suppressed compared to the EOD based on the neat HDT-1 film because of the HDT-1's deep LUMO level to limit the electron transport due to the electron traps in the codeposited films. The reason for the variation in the electron transport properties of the codeposited films would originate from molecular packing. The calculated HOMO and LUMO distributions of the host and the guest molecules are shown in Supplementary Fig. 3. Although the HOMO is widely distributed through the host molecules because of the donor Cz moieties, the LUMO distributions are relatively located on the center of the molecules. Thus, because large intermolecular LUMO overlaps are necessary for efficient intermolecular electron transfer, sufficient molecular packing is beneficial to enhance the electron transport in amorphous films[26]. Therefore, host molecules with large molecular sizes, such as 2DPCz and 3DPCz with the bulky donor Czs, have a slight disadvantages for efficient electron transport in the codeposited films.

The OLED structure is composed of indium-tin-oxide (ITO; 100 nm)/1,4,5,8,9,11-hexaazatriphenylene hexacarbonitrile (HAT-CN; 10 nm)/N,N′-di(1-naphthyl)-N,N′-diphenyl-(1,1′-biphenyl)−4,4′-diamine (NPD; 40 nm)/tris(4-carbazoyl-9-ylphenyl)amine (TCTA; 15 nm)/EML (30 nm)/2-(9,9′-spirobi[fluoren]−6-yl)−4,6-diphenyl-1,3,5-triazine (SF3-TRZ; 5 nm)/50 wt% 8-quinolinolato lithium (Liq): SF3-TRZ (30 nm)/Liq (2 nm)/Al (100 nm) as shown in Fig. 2d, and the chemical structures are summarized in Supplementary Fig. 4. Here, the EML is composed of a codeposited film of 10 mol% HDT-1 doped in the different hosts. The J-V-luminance (L) characteristics of the TADF-OLEDs are shown in Fig. 2e and Supplementary Fig. 5. The device based on the 1DPCz host showed the best J-V-L characteristics compared to the other devices. Based on the comparison of the HOD and EOD characteristics with

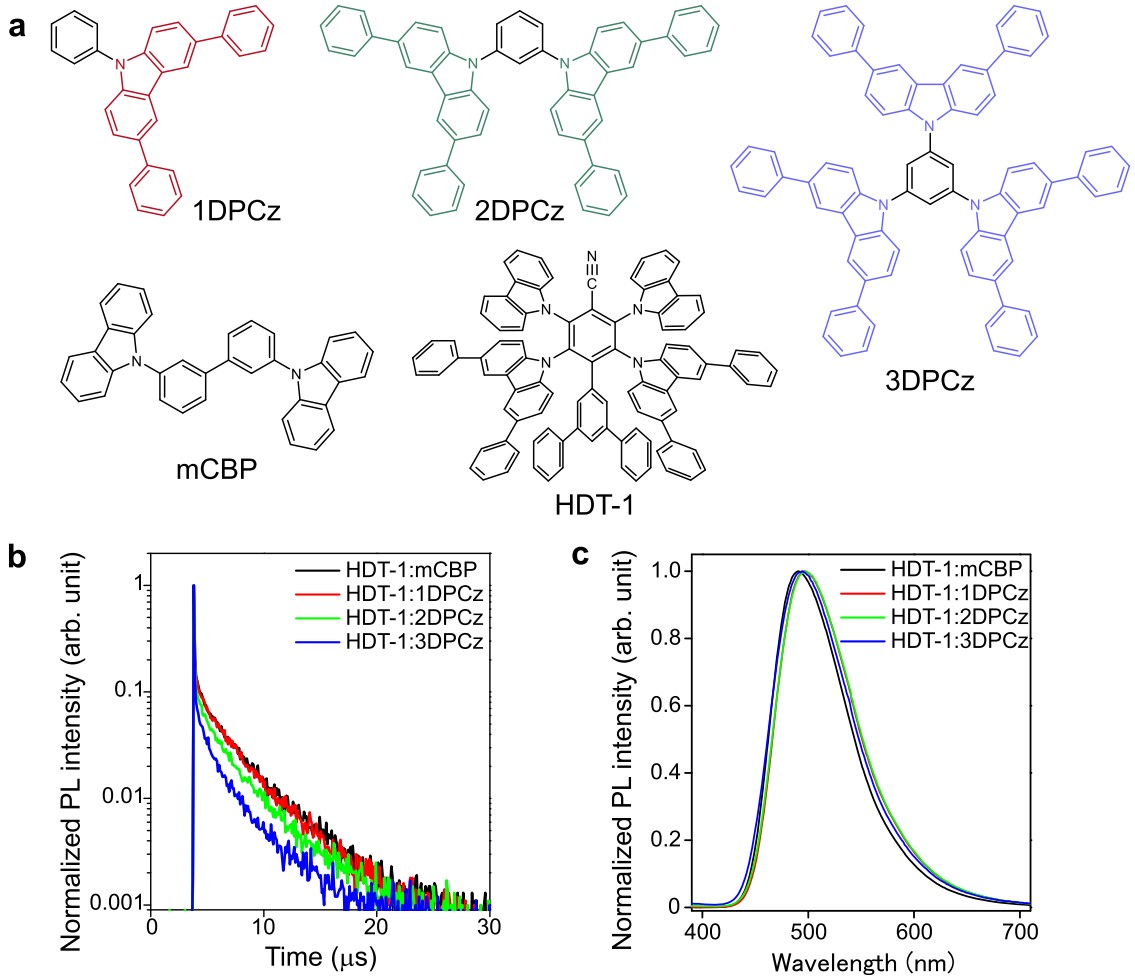

**Fig. 1 | Carbazole-based host molecules and photophysical properties of codeposited films of TADF emitter host molecules. a** Molecular structures of TADF emitter and host molecules. **b** Transient photoluminescence (PL) decay profiles of codeposited films. **c** PL spectra of codeposited films.

mCBP and 1DPCz hosts, this can be attributable to the enhancement of the hole transport ability in the 1DPCz-based EML (Fig. 2a). On the other hand, the turn-on voltage (the voltage corresponding to $L = 1$ cd m$^{-2}$) of the 2DPCz-based OLED is higher than those of others. Although the 3DPCz host exhibited relatively high driving voltage in both HOD and EOD compared to those of the 2DPCz host (Fig. 2a, b), the turn-on voltage in the 2DPCz-based OLED is higher than that of the 3DPCz-based OLED (Fig. 2e).

To explain the inconsistency of the difference in turn-on voltage in these OLEDs, we investigated the effect of EML polarization on carrier injection into EMLs. The EML polarization, i.e., GSP, was investigated by surface potential measurement for sequentially deposited films using a Kelvin probe method[8,10]. The calculated PDM direction of the TADF and the host molecules is depicted in Supplementary Fig. 6. Figure 3a shows the thickness dependence of the surface potential for the deposited neat films of each host matrix and HDT-1 molecules. Their GSP slope values are summarized in Supplementary Table 1. Although a neat HDT-1 film showed a positive GSP (+7.0 mV nm$^{-1}$), an mCBP and a 2DPCz films showed a negative GSP of −5.3 mV nm$^{-1}$ and −47 mV nm$^{-1}$, respectively. Since a 3DPCz molecule possesses no PDM due to the symmetrical molecular structure, the GSP slope in the 3DPCz film was only +2.2 mV nm$^{-1}$, i.e., nearly zero. A 1DPCz film also showed a relatively small GSP slope (−1.5 mV nm$^{-1}$) compared to those of mCBP and 2DPCz, despite the relatively large PDM magnitude of 2.1 Debye (Supplementary Fig. 6).

The surface potential of the codeposited films of 10 mol% HDT-1 doped in various host matrices was then measured (Fig. 3b and

Supplementary Table 1). The nearly nonpolar 3DPCz host increased the GSP slope of the HDT-1-doped film (+12 mV nm$^{-1}$) compared to that of the neat HDT-1 film despite the reduced number of polar molecules in the film. This can be attributed to the suppression of anti-parallel dipole-pair formation that is induced by the intermolecular dipole-dipole interactions between polar HDT-1 molecules. The nonpolar host molecules can act as a spacer between the polar molecules and suppress intermolecular dipole-dipole interactions, leading to the formation of the parallel dipole orientation[9,22,27]. Thus, the PDM orientation degree of the HDT-1 molecules is improved in the 3DPCz host matrix (Fig. 3c). Doping HDT-1 into the host matrices having negative GSP slopes, such as mCBP and 2DPCz, induces the positive shift of GSP slope values compared to the neat host films (Supplementary Table 1) because of the positive and strong PDM orientation of HDT-1 (Fig. 3c). Interestingly, the 1DPCz:HDT-1 codeposited film showed nearly no thickness dependence of GSP, i.e., a very small GSP slope = +1.5 mV nm$^{-1}$. This result indicates that the 1DPCz molecules cancel out the net polarization induced by the SOP of HDT-1 molecules in the matrix, leading to the formation of a nonpolar film.

We suppose that the cancellation of the SOP by the 1DPCz matrix originates from the high molecular diffusivities of 1DPCz molecules, inducing the random orientation. It has been pointed out that the ratio of substrate temperature ($T_s$) and glass transition temperature ($T_g$), i.e., $T_s/T_g$, is one of the indicators correlating to the molecular diffusivity of molecules on a substrate during a vacuum deposition process. Therefore, we can expect high molecular diffusivities for a low $T_g$ molecule and vice versa at a constant $T_s$ value[21,27–30]. Previous studies

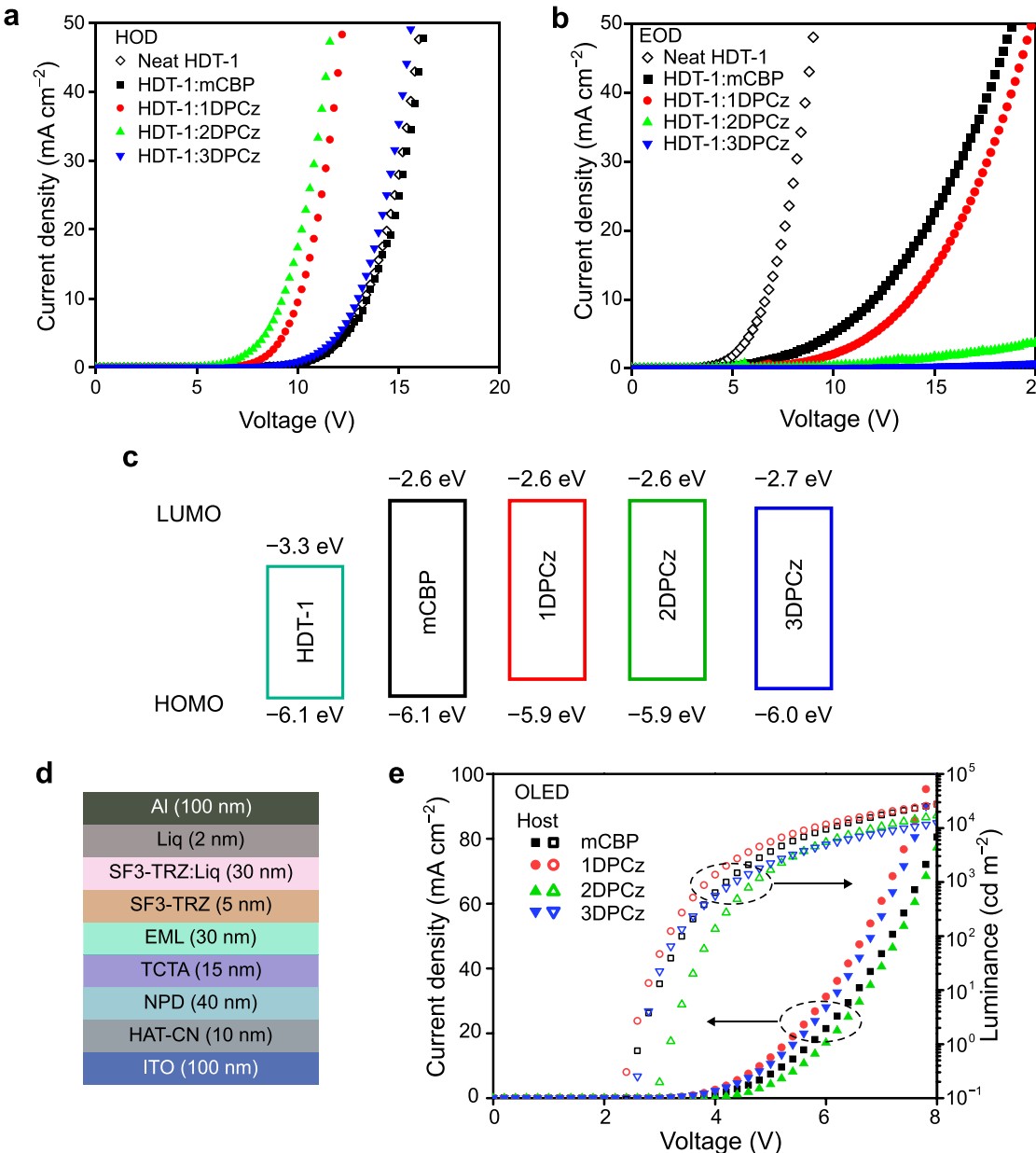

**Fig. 2 | Carrier transport properties of codeposited films. a** Current density-voltage characteristics of hole-only devices. **b** Current density-voltage characteristics of electron-only devices. **c** Highest occupied molecular orbital (HOMO)-lowest unoccupied molecular orbital (LUMO) energy diagrams. **d** Device structure of OLEDs. **e** Current density-luminance-voltage characteristics of OLEDs.

about the orientation of transition dipole moment (TDM) of a deposited molecule revealed that a high $T_g$ induces a highly horizontal TDM orientation of a molecule, while a low $T_g$ molecule randomizes the molecular orientation[21,28,30], indicating that excess molecular diffusivity induces the random orientation of the deposited molecules. The $T_g$s of 1DPCz, 2DPCz, and mCBP were found to be 60, 140, and 90 °C, respectively (Supplementary Table 2). Note that the $T_g$ of 3DPCz was not clearly observed due to the high melting point which is close to the decomposition temperature. Thus, the low $T_g$ of 1DPCz enabled the random PDM orientation in the neat film, leading to the almost zero GSP (Fig. 3a). Similarly, the 1DPCz molecules form the nearly-random orientation in a codeposited film with HDT-1. Since 1DPCz possesses high diffusivities due to the small molecular size, the 1DPCz molecules effectively cancel out the polarization of the HDT-1 (Fig. 3c). Further, 2DPCz is also a polar molecule with a comparable PDM magnitude to

1DPCz. However, the high $T_g$ of 2DPCz, i.e., low molecular mobility, inhibits the canceling of the SOP in the HDT-1:2DPCz film.

To reinforce our claim, we additionally evaluated the SOP characteristics of an HDT-1 doped nonpolar 4,4'-di(9H-carbazol-9-yl)-1,1'-biphenyl (CBP) matrix having a comparable $T_g$ (62 °C)[31] to 1DPCz. The HDT-1:CBP film showed a positive GSP slope (+7.8 mV nm⁻¹) (Supplementary Fig. 7), meaning that a nonpolar CBP matrix could not cancel out the SOP of HDT-1 while the CBP molecules showed the random orientation in the film[9]. Therefore, both the low $T_g$ and moderate PDM magnitude are ones of the essential factors for host molecules to form a nonpolar codeposited film. In order to prove it, the order parameter (S) of a TDM orientation of HDT-1 in different host matrices was determined by an angular-dependent PL profile[32]. The S value is given by $S = \frac{1}{2}\langle 3\cos^2\theta - 1 \rangle$, where $\theta$ is the angle between the TDM and the substrate normal. Here, S value of 0 and −0.5 corresponds to a random

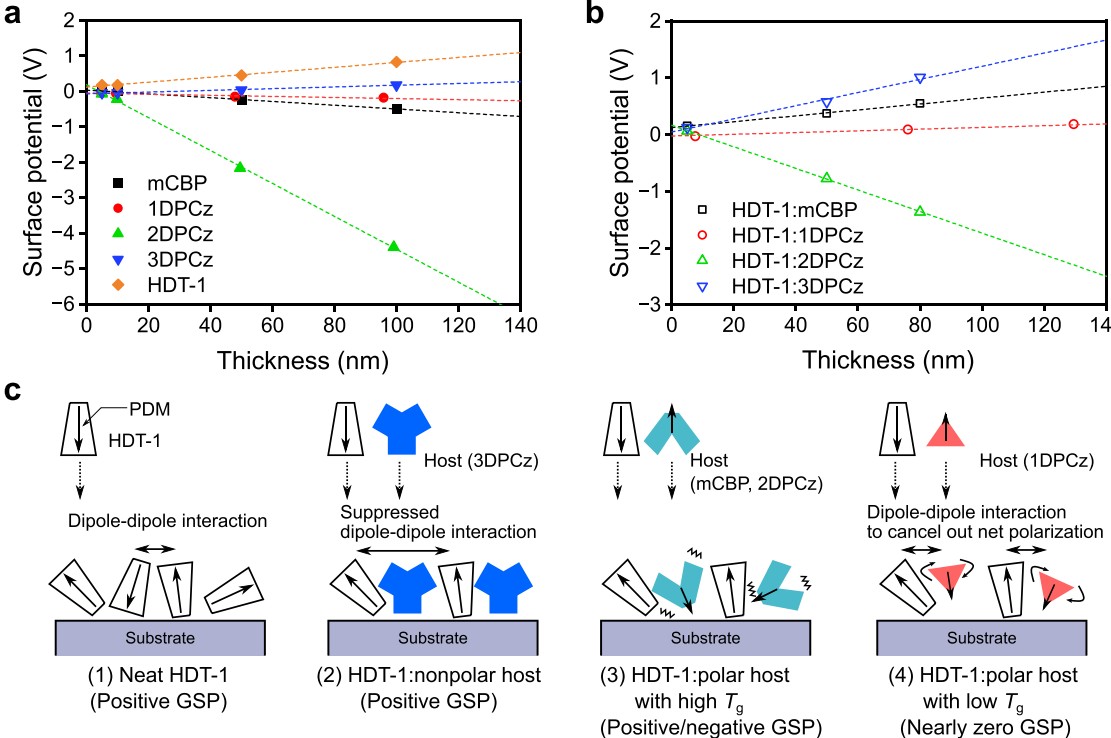

**Fig. 3 | Spontaneous polarization formation of vacuum-deposited films.** Thickness dependence of the surface potential of neat films (**a**) and codeposited films (**b**) of HDT-1 and host molecules. The dashed lines show the fit lines. **c** Schematics of spontaneous polarization formation of the combination of HDT-1 and several host molecules.

and a horizontal TDM orientation, respectively. The codeposited films of HDT-1 with an mCBP or a 1DPCz host exhibited similar angle dependence of the PL intensity (Supplementary Fig. 8a), indicating that the HDT-1's TDMs in the 1DPCz host matrix form comparable horizontal orientation to the codeposited film with the mCBP, i.e., $S = -0.41$[23]. Note that we estimate that the TDM direction of HDT-1 is oriented parallel to the central phenyl ring (Supplementary Fig. 8b). Thus, we concluded that the non-polarization of the HDT-1:1DPCz film is induced by the polarization cancellation of 1DPCz molecules, not the change in the orientation of HDT-1. This would benefit to form a nonpolar EML without randomizing a horizontal TDM orientation of emitter molecules, i.e., high light-outcoupling efficiency can be maintained in OLEDs.

The film polarization in these films was further confirmed by a displacement current measurement (DCM)[33] using a bilayer device structure (Supplementary Fig. 9a). We note that the measured displace current is proportional to the capacitance of the devices, and the DCM profiles mainly correspond to the hole carrier behavior due to no electron injection layer in the tested device structures (Supplementary Fig. 9b–d). The DCM profiles of the devices based on the codeposited films with positive GSP, such as HDT-1 with mCBP or 3DPCz, exhibited clear terrace characteristics in the DCM profiles (Supplementary Fig. 9e). This profile indicates that the positive GSP of the films induces hole injection from the ITO electrode and the successive hole accumulation at the interface between the NPD and the codeposited layers under the applied voltage below the threshold voltage of the actual current. In contrast, the 1DPCz-based device exhibited no carrier accumulation behavior in the DCM profile, which is similar results of a device based on a nonpolar CBP layer (Supplementary Fig. 9f). Thus, the HDT-1:1DPCz film could be assigned to a nonpolar film as investigated by the Kelvin probe method. The previous study regarding the influence of GSP polarity on charge injection for OLEDs[10] pointed out that the organic layer exhibiting negative GSP results in low carrier injection efficiency. This is because the carrier injection is suppressed

by the positive and the negative interface charges at the bottom- and the top-side interfaces. Here, we confirmed that the HDT-1:2DPCz EML suppresses carrier injections, leading to a deterioration of the turn-on voltage in the 2DPCz-based OLED (Fig. 2d). We note that since the GSP slopes of the SF3-TRZ and the SF3-TRZ:Liq codeposited films are small compared to the EMLs in this study (Supplementary Fig. 10)[27], the SOP of the EMLs mostly affect the carrier injection properties of the OLEDs.

To demonstrate the wide applicability of non-polarization by codeposition with 1DPCz, another polar TADF molecule, 1,2,3,4-tetra-kis(carbazol-9-yl)-5,6-dicyanobenzene (4CzPN), was also tested. Supplementary Fig. 11 shows the thickness dependence of surface potential of a neat 4CzPN and a 10 mol% 4CzPN:host films. The GSP slope values of the neat 4CzPN film and the codeposited film with 1DPCz were +55 mV nm⁻¹ and +1.6 mV nm⁻¹, respectively (Supplementary Table 3). These results clearly indicate that the 1DPCz's ability to cancel out SOP is widely applicable as an EML host. In the case of the 2DPCz and 3DPCz hosts, the SOP polarities were the same as the films with HDT-1, i.e., the 4CzPN:2DPCz film and the 4CzPN:3DPCz films showed negative and positive SOPs. The *J-V* characteristics (Supplementary Fig. 12) of the 4CzPN-OLED with the 2DPCz host also showed higher turn-on voltage derived from the negative SOP of the codeposited film (see Supplementary Note 1).

Figure 4a shows the EQE-*J* characteristics of the OLEDs based on each host matrix. The 1DPCz-based OLED exhibited a high maximum EQE ($EQE_{max}$ = 26.4%) compared to the mCBP-based OLED ($EQE_{max}$ = 25.3%), while PLQY = 90% of an HDT-1:1DPCz film is lower than PLQY = 95% of an HDT-1:mCBP film. The higher EQE can be attributed to the improved hole transport, resulting in a better carrier balance and/or suppressed exciton quenching by accumulated interfacial charges[11,13,19,34]. We note that these devices were fabricated in the same batch using the method of replacing several shadow masks for the EMLs, meaning that the organic layers, except for the EMLs in each device, are identical. Further, the relatively small batch-to-batch

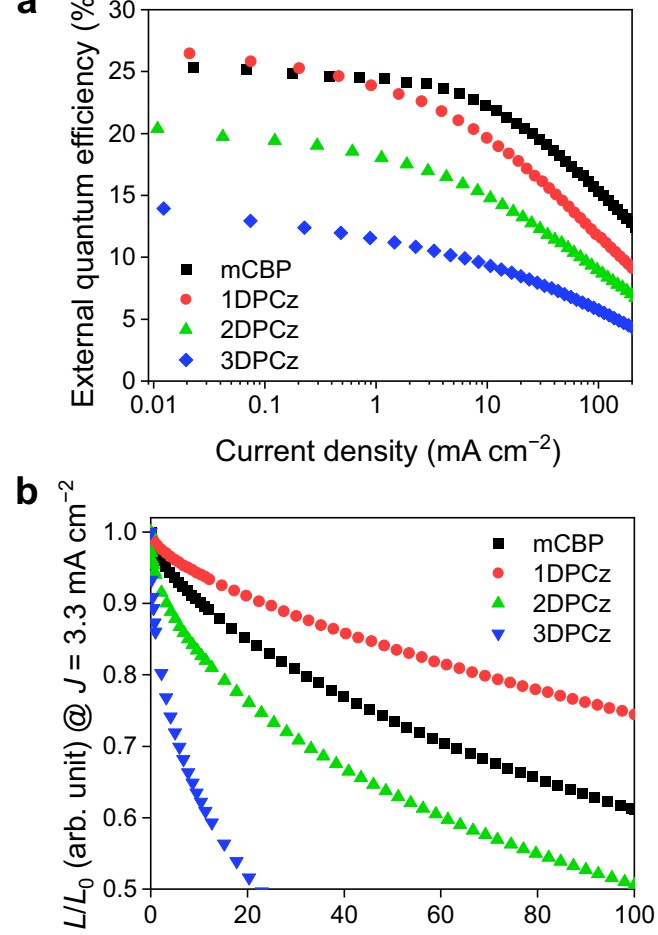

**Fig. 4 | Device performance of TADF-OLEDs. a** External quantum efficiency-current density (J) profiles of TADF-OLEDs based on HDT-1. **b** Relative luminance (L) decay to initial luminance ($L_0$: initial luminance) measured under constant current operation (J: 3.3 mA cm⁻²).

variation of EQE was also confirmed (Supplementary Fig. 13). Thus, we can identify the small EQE difference as a significant issue. Previously, we revealed that a blueshift of EL spectra with an increase in current density results from the shift in a recombination zone profile toward the cathode-side in TADF-OLEDs with an mCBP host[7]. Thus, the high electron-transport property of mCBP induces carrier recombination near the HTL/EML interface at a low current density region. Here, a similar EL blueshift was confirmed in the mCBP- and the 1DPCz-based OLEDs in this study (Supplementary Fig. 14). Further, the mCBP-based OLED showed a large spectral shift compared to the 1DPCz-based OLED, indicating that the recombination zone of the mCBP-based EML is more concentrated near the HTL/EML interface at a low current density. In contrast, the enhanced hole transport in the 1DPCz-based EML improves the carrier balance compared to the mCBP-based EML. The carrier recombination zone in the 1DPCz-based OLED, therefore, shifts toward the electron-transport layer (ETL)-side, suppressing carrier and exciton accumulation at the HTL/EML interface. While the effect of GSP-induced accumulation charges on exciton quenching is mostly observed under a voltage lower than the turn-on voltage, previous studies have reported that the exciton quenching by the accumulated charges limits the internal quantum efficiency at the low current density less than 10⁻¹ mA cm⁻², meaning that the GSP effect influences the shape of the EQE shape[13,19]. The estimated accumulation

charge density at the interface of the mCBP-based EML is 0.14 mC m⁻². Although this value is relatively small compared to conventional SOP molecules such as Alq₃ and TPBi, the EML with small polarizations would decrease IQE because of the presence of negative and positive polarization charges at both interfaces of the EML. As a result, the nonpolar 1DPCz-EML can avoid unwanted exciton interactions such as triplet-triplet and triplet-polaron annihilations[5,6,11], leading to a higher EQE_max. However, the 1DPCz-based OLED exhibited severe EQE rolloff at a high current region compared to the mCBP-based OLED. As discussed above, this OLED structure often induces the shift of the recombination zone toward the ETL side at a high current density. Therefore, the EQE rolloff would originate from the exciton annihilations near the EML/ETL interface, which is consistent with the higher hole-transport property of the 1DPCz-based EML. Further, the severe rolloff can be suppressed by the improvement of electron transport and injection properties of the ETL-side materials.

The EQE performances of OLEDs based on the 4CzPN:host EMLs with larger differences in the SOP are shown in Supplementary Fig. 15a. The 4CzPN-based OLEDs with different hosts exhibited comparable EQE values because of the similar PLQY (34-38%) of the codeposited films. The effects of the EPQ caused by the SOP-induced accumulated charges were investigated by a DCM and a bias-dependent PL measurements (Supplementary Fig. 15b–f and Supplementary Note 2). We confirmed a quite small decrease in the PL intensity under the voltage range from the carrier accumulation voltage to the actual current injection voltage, and estimated that this PL decrease originates from the exciton quenching by the accumulated holes at the EML interface. The reason for the quite small change in the PL intensity is not only the small amount of the quenched excitons but also the thicker EML than that of the previous studies, resulting in a decrease in the percentage of the photogenerated excitons quenched at the limited quenching site near the interface (Supplementary Fig. 16). Furthermore because a relatively long exciton lifetime (μs-ms) of TADF emitters is strongly influenced by a voltage applied across the film[35], a shape of the bias-dependent PL profiles contains a portion of exciton quenching via an applied electric filed (Supplementary Fig. 15c–e)[36]. Thus, it might be necessary to apply some techniques for the precise identification of the percentage of SOP-induced exciton quenching in TADF-based OLEDs by the bias-dependent PL measurement.

Recent research revealed that the reduction of SOP in ETLs clearly improves device stability under continuous driving conditions since the exciton annihilations, such as interfacial EPQ, are suppressed in the device with a nonpolar ETL[15,16]. Indeed, the EML polarization would also strongly affect the device stability. Figure 4b shows the luminance decay profiles of the HDT-1-based OLEDs under continuous device operation at constant current density (J = 3.3 mA cm⁻²). Although the 2DPCz- and the 3DPCz-based devices showed quick degradation compared to the conventional mCBP-based device, the 1DPCz device exhibited improved operational stability. The trend of the device stability was confirmed in the case of the measurement with the same initial luminance ($L_0$) of 1000 cd m⁻² (Supplementary Fig. 17). Besides, the 4CzPN-based OLEDs with the 1DPCz-host exhibited improved device stability compared to the OLED with 2DPCz (Supplementary Fig. 18). We estimate that one of the reasons for the improved device stabilities is the small polarization of the EML based on a 1DPCz host. In the case of a positive EML polarization, injected holes and electrons accumulate at the both of TCTA/EML and the EML/SF3-TRZ interfaces, respectively. On the other hands, a negative polarization of EMLs induces holes and electrons accumulation at the both of EML/SF3-TRZ and the TCTA/EML interfaces, respectively. Previous studies on the impact of SOP in OLED stability have focused on positive SOP of ETLs, which induces the hole accumulation at the EML/ETL interface[10,11,13,15,19,37]. In contrast, the SOP of EMLs cause charge accumulations (hole and electron) at both interfaces, which would critically affect the device stability because of the severe EPQ-driven

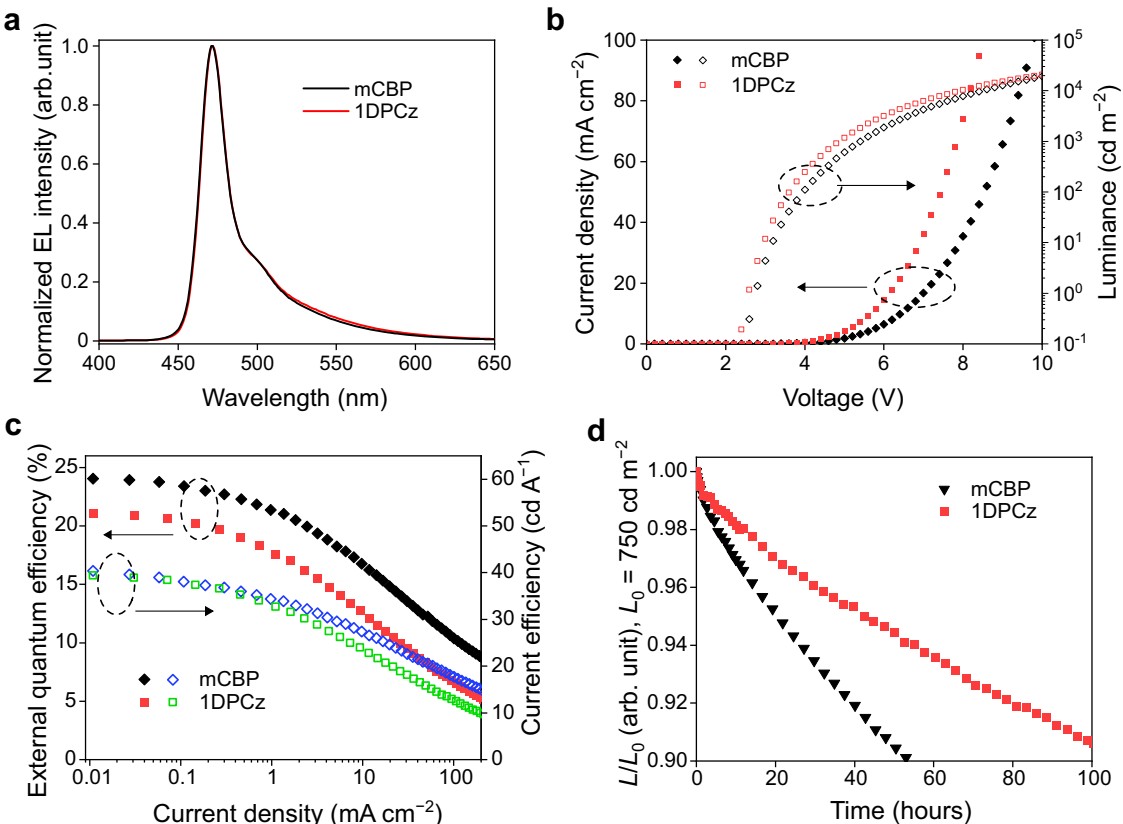

**Fig. 5 | Device performance of TAF-OLEDs. a** Electroluminescence (EL) spectra. **b** Current density-luminance-voltage characteristics, **(c)** External quantum efficiency and current efficiency profiles. **d** Relative luminance ($L$) decay to initial luminance ($L_0$) measured under constant current density ($L_0$: 750 cd m$^{-2}$).

degradations at both interfaces of the EML. Although we could not clearly observe the interfacial charge accumulations (especially for the 2DPCz-based EMLs with the negative polarizations) and the exciton quenching through EPQ by the DCM and the bias-dependent PL measurement, a little effect of the SOP results in the negative impact on the device stability due to continuous long-time device operations (Supplementary Note 3 and Supplementary Fig. 19). Another factor affecting the device stability is the carrier transport properties of the EMLs. The balanced carrier transports suppress EPQ, leading to longer device lifetimes. As discussed above, the HDT-1:1DPCz-EML possesses a better carrier balance due to the high hole transport and the moderate electron transport properties, reducing the charge accumulation near the EML interfaces. Thus, we estimate that both of the reduction in SOP and the improved carrier balance result in the suppression of the undesirable triplet-polaron and triplet-triplet annihilations, leading to the generation of high-energy excitons and/or polarons acting as a degradation source[5,6,11].

Finally, to exploit the full potential of the HDT-1:1DPCz-based blue OLED, we fabricated TADF-assisted fluorescence (TAF) OLEDs[38]. The device structure was ITO (50 nm)/HAT-CN (10 nm)/NPD (40 nm)/TCTA (10 nm)/1DPCz (5 nm)/EML (40 nm)/SF3-TRZ:Liq (25 nm)/Liq (2 nm)/Al (100 nm), where the EML comprises of the hosts (mCBP and 1DPCz), 25 wt%-doped HDT-1 as a triplet sensitizer, and 0.5 wt%-doped v-DABNA as a blue-emitter molecule (Supplementary Fig. 20)[39]. The EL spectra (Fig. 5a) of the TAF-devices showed narrowed blue EL with full-width at half-maximum of 20 nm (CIE ($x$, $y$) coordinates of (0.15, 0.26)) that originates from v-DABNA, indicating the occurrence of an efficient Förster energy transfer from HDT-1 to v-DABNA[23]. The $J$-$V$ characteristics shown in Fig. 5b were dramatically improved in the 1DPCz-based TAF-OLED due to the well-balanced carrier transport properties of 1DPCz. Although the EQE$_{max}$ of the 1DPCz-based TAF-OLED slightly decreased (Fig. 5c) due to the shift of the carrier recombination site, the

operational device stability (LT$_{95}$) of the 1DPCz-based TAF-OLED was 42 hours that is two times longer than that of the mCBP-host device as shown in Fig. 5d ($L_0$: 750 cd m$^{-2}$). Thus, the 1DPCz-based TAF-OLED with low $T_g$ provided high device stability compared to other devices with high $T_g$ hosts such as mCBP. This indicates that the control of EML polarization and carrier balance is more crucial to achieving stable device operation than developing a high $T_g$ host molecule. While developing high $T_g$ hosts, in fact, is one of the important criteria for the practical application of OLEDs[4], the simultaneous achievement of a high $T_g$ matrix and the cancellation of the film polarization will be possible using mixed host strategies[40]. Indeed, the TAF-OLED with the 1DPCz:mCBP mixed host exhibited higher device stability compared to the single-host OLEDs (Supplementary Fig. 21). This indicates that a fine-tuning of EML properties such as film polarization, carrier balance, and $T_g$ critically surely improves the OLED performance.

In conclusion, we revealed that host molecules play a critical role as a regulator in tuning the EML polarization which induces unwanted carrier accumulation at the interface. A polar host molecule with low $T_g$ can reduce the polarization of a codeposited film with polar emitter molecules. The 1DPCz host molecules cancel out the film polarization induced by the HDT-1's PDM orientation due to the high molecular mobility and the moderate PDM. Optimized carrier transport and cancellation of the film polarization clearly elongated device stability of the TADF- and TAF-OLEDs. Thus, our molecular design and selection strategy of a host molecule to control the film polarization allow emitter molecules to exhibit their underlying potential, leading to improved OLED efficiency and stability.

## Methods
### Materials
Organic semiconductor molecules for the tested OLEDs were synthesized in-house, and purified by temperature gradient vacuum

sublimation[24]. Differential scanning calorimetry analysis was performed by Netzsch DSC204 Phoenix calorimeter at a scanning rate of $10\,°C\,min^{-1}$ under an $N_2$ atmosphere. Optimized molecular structures and permanent dipoles of the ground state molecules were calculated via DFT calculations at the B3LYP/6-31 G (d) level with the Gaussian 16 program package. Transition dipole of HDT-1 was calculated via a TD-DFT calculation at the CAM-B3LYP/6-31 G (d) level of the Gaussian 16 program package.

### Sample fabrication
OLEDs and bilayer devices for DCM were fabricated by vacuum vapor deposition processes without exposure to ambient air. After fabrication, the devices were immediately encapsulated under a glass cover using epoxy glue in a nitrogen-filled glovebox ($H_2O > 0.1$ ppm, $O_2 > 0.1$ ppm). All organic layers, except for the EML and the Liq layer, were deposited at a deposition rate of $0.1\,nm\,s^{-1}$, whereas the EML and the Liq layer were deposited at 0.05 and $0.04\,nm\,s^{-1}$, respectively. The deposition rate of Al was $0.2\,nm\,s^{-1}$. The device area was $0.04\,cm^2$.

### Photophysical characterization
UV–vis absorption and PL spectra were recorded on a Lambda 950 KPA spectrophotometer (PerkinElmer, US) and FP-8600 PL spectrometer (JASCO, Japan). Phosphorescent spectra were recorded on an FP-8600 spectrometer at $-196\,°C$. The PLQY values were measured using a Quantaurus-QY system (C11347-11, Hamamatsu Photonics) in flowing argon gas at an excitation wavelength of 300 nm. The transient PL decay profiles were measured using Quantaurus-Tau system (C11367-25, Hamamatsu Photonics). To determine a HOMO energy level, the photoemission yields were measured on an AC-3E (Riken Keiki, Japan). A LUMO energy level was determined from (HOMO energy level) − (absorption edge energy). A $T_1$ level was determined from the onset energy of a phosphorescent spectrum. To evaluate the dipole orientations of the emitter in doped films, 15-nm-thick doped films were deposited on glass slides (refractive index $n = 1.52$). Angular-dependent PL measurements were performed using the C14234-01 measurement system (Hamamatsu Photonics, Japan) at an excitation wavelength of 365 nm and the other parameters were determined by fitting the profiles using Setfos 4.6 software (Fluxim, Switzerland).

### Surface potential measurement
Organic solid-state films with various thicknesses for surface potential measurement were directly deposited on a pre-cleaned 100-nm-thick ITO-coated glass substrate. The vacuum deposition was performed under a high vacuum at pressure levels below $1 \times 10^{-4}$ Pa with a monitoring deposition rate using a house-made evaporation machine. The deposition rate was controlled at $0.05\,nm\,s^{-1}$. The surface potential was measured using the Kelvin probe method under a vacuum condition of the pressure level below $1 \times 10^{-4}$ Pa. The grown organic film was directly transferred from the evaporation chamber to the measurement chamber without air and light exposure. The surface potential of the organic film surface was then measured as a function of film thickness by using a Kelvin probe (UHVKP020, K.P. Technology Ltd) with reference to the ITO electrode under a dark condition. The total film thickness was estimated after the surface potential measurement using a stylus profilometer (DEKTAK-XT, BRUKER).

### Device characterization
The EQE and $J-V-L$ measurements were performed using a calibrated luminance meter (CS-2000, Konica Minolta). EQE values were collected by the results of angular-dependent EL profiles (C9920-11, Hamamatsu Photonics). For the device lifetime tests, the luminance and EL spectra of the driving devices in the normal direction were measured using a luminance meter (SR-3AR, Topcon) at a controlled temperature of 30 °C. For the DCM, repeated-triangular voltage signals were applied to each device using a function generator (WaveStation 2052, Teledyne LeCroy), and the displacement current was amplified using a current amplifier (CA5350, NF electronic instruments). The applied voltage and the amplified current were measured using an oscilloscope (HDO4054A, Teledyne Lecroy). The applied voltage scan rate was $100\,V\,s^{-1}$.

## Data availability
The experimental data in this study are provided in the Source Data file. Source data are provided with this paper.

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

## Acknowledgements

The authors thank Dr. Hiroki Noda, Ms. Keiko Kusuhara, and Ms. Nozomi Nakamura of Kyushu University for preparing the chemicals and their thermal analysis. The authors also acknowledge Dr. Hiroshi Fujimoto, Dr. Hin-Wai Mo, and Ms. Kaori Nagayoshi from i3-opera for their help with sample fabrication. This work was supported in part by the Program for Building Regional Innovation Ecosystems of the Ministry of Education, Culture, Sports, Science and Technology (MEXT), Japan Society for the Promotion of Science (JSPS) KAKENHI (JP23H05406 (C.A.) and JP23K13716 (M.T.)), Japan Science and Technology Agency (JST) FOREST Program (JPMJFR223S (M.T.)), and Inamori Foundation (M.T.).

## Author contributions

The project was conceived and designed by M.T. M.T. prepared the films and the devices and measured their properties. C.-Y.C. synthesized the TADF and the host molecules. M.T., C.-Y.C., H.N. and C.A. contributed to writing the paper and critically commented on the project.

## Competing interests

The authors declare no competing interests.
