## [Peer Review File · Nature Communications]

Simultaneous control of carrier transport and film polarization of emission layers aimed at high-performance OLEDsREVIEWERS' COMMENTS:

Reviewer #1 (Remarks to the Author):

This work by Tanaka et al. addresses film polarization by orientation of polar molecules in organic layers and its effect on OLED efficiency and lifetime. Specifically, the authors have engineered the emissive guest-host system consisting of a polar TADF emitter and host materials with different polarization. Surprisingly they find that the combination of the polar emitter with the host having the highest molecular dipole moment and the lowest glass transition temperature results in the longest-lived OLEDs. This is remarkable and not expected from the conventional view of OLED stability requiring high-T_g materials.

However, one has to be aware that the observed differences are not tremendously big and that it is not straightforward to pin them down to a single origin. To be more precise: OLED efficiency and lifetime are depending on several factors that need to be well balanced. Here the authors mainly focus on polarization of the emission layer, which will result in charge accumulation at the interface to the hole (electron) transport layer. This kind of charge pile-up is one source of efficiency loss, but it is effective mostly at low voltage or even below turn on of the OLED. What is more relevant in the range where OLEDs are typically operated, is charge distribution and the resulting recombination profile due to injected charges in the EML. These two effects are not clearly distinguished in the paper.

The GSP variations between different hosts are anyway small -- except for 2DPCz. Thus, it is hard to believe that GSP is the only factor that is responsible for the longer lifetime of the OLED with 1DPCz as host. It could well be an interplay of several factors, like energy levels, charge mobility, etc., which leads to a different charge distribution in the OLED EML.

I think one would need to do a more detailed analysis, including e.g. device simulation and emission zone analysis to pin down the origin of device improvements.

Additionally, I have spotted some other points that need clarification:

- The sentence "However, neither molecular design nor deposition method has been reported to actively control SOP." in the abstract is not justified. There have been several reports on the control of SOP, and some of them are cited in the paper as well (e.g. 13, 17, 20 ...).
- Single carrier devices are not analyzed in much detail. What about SCLC and carrier

mobilities?

- It doesn't make sense to use different scaling for the vertical axes of IVL characteristics -- one being linear and the other logarithmic, although this is often done in the literature. It is impossible to correlate one property (I) with the other (L) in such graphs.
- I don't understand the argument with the different kinetic energy of molecules having different Tg's. The kinetic energy of molecules on a surface is given by the substrate temperature and thus $E_{kin} \sim k_B * T_S$ (or the like) is independent of the Tg. What the authors actually mean is surface diffusivity, I guess.
- Correlation between PDM and TDM alignment, where the latter is taken from angular PL measurements, is not a straightforward task. One needs to know the relation between them on the molecules, as obtained e.g. by DFT calculations. The authors have only shown the PDM calculations in their manuscript.
- Differences in EQE between the mCBP and 1DPCz hosts are rather small. I am wondering how large the error bars of the single measurements are and how the statistics of several nominally identical OLED devices is.

Reviewer #2 (Remarks to the Author):

The authors report a strategy to achieve high operational stability of OLEDs by reducing spontaneous orientation polarization (SOP) and improving carrier balance in the emission layer (EML). They found 1DPCz, which has a high molecular kinetic energy and moderate permanent dipole moment (PDM), efficiently eliminates the total SOP in the EML incorporating a polar TADF emitter. The hole conductance of this material is higher than that of the reference material, mCBP, while the electron conductance is comparable. The operational stability as well as the maximum EQE of the OLED are improved by using 1DPCz as a host material, even though Tg of this material is relatively low. They further applied this molecule to the host of a blue TAF-OLED, and demonstrated its high operational stability. The results presented in this manuscript are interesting and would benefit the OLED community; however, the underlying mechanisms are still ambiguous (see comments below). Moreover, a similar work has been published recently, so that the impacts of this work would be moderate. Overall, the reviewer considers that this work does not meet the high standard of Nature Communications.

1. A similar work has been recently published (E. Afolayan et al., Phys. Rev. Applied, 10.1103/PhysRevApplied.17.L051002). The authors should refer to this paper and state the significance and novelty of this work.

2. (Fig. 4 and Fig. 5, p.10 and p.11) The authors discussed the origin of improvement of the device performance; however, the influence of reducing SOP on the device performance is still a speculation. The authors should estimate the contributions of the carrier balance (or recombination zone) and SOP to the improvements of EQE and lifetime, quantitatively. For instance, although the GSP slope of the 1DPCz:HDT-1 film is very small (1.5 mV nm^{-1}), that of the mCBP:HDT-1 film is also small (5.2 mV nm^{-1}). The reviewer is wondering whether this difference (corresponding to $\sim 0.1 \text{ mC m}^{-2}$) can lead to the higher EQE (Fig. 4a) and the significant improvement of the operational stability (Fig. 4b, Fig. 5d).

3. The SOP of ETL often facilitates the charge accumulation at the EML/ETL interface. The authors should also examine the GSP of SF3-TRZ and SF3-TRZ:Liq films.

4. The title of the paper should be reconsidered. Although the enhancement of the lifetime of a blue OLED is demonstrated, the results and discussions about the "blue OLED" are rather additional (as the authors indicate in the introduction). Further, the contribution of "film polarization" to the operational stability is still unclear (as suggested in the second comment).

5. The improvement of the carrier balance in EML is also suggested as an important factor, in terms of the host selection strategy. The authors should discuss the origin of the enhanced hole conductivity in the HDT-1:1DPCz(2DPCz) film as well as the reduced electron conductivity in the HDT-1:2DPCz(3DPCz).

6. Ref. 11 and 32, 19 and 27 are the same.

7. (Figure 1 caption) HDT-a should be HDT-1.

**Reply to the comments**

Firstly, we sincerely thank all the reviewers for taking her/his precious time to evaluate
our manuscript and providing insightful and helpful comments and suggestions. These
are keys to helping us to improve the quality of our manuscript. Our original manuscript
lacked the essential experimental data to show the universality of our proposed molecular
designs, leading to the reviewers' confusion. Although the film polarization of HDT-1 in
the original manuscript is relatively small, we also evaluated the codeposited film's
polarization properties with a 4CzPN guest molecule exhibiting a strong polarization. The
newly designed host molecule in this manuscript, 1DPCz, successfully canceled out the
4CzPN polarization to form a nonpolar emission layer, indicating the universality of the
1DPCz's ability. Furthermore, we agree that carrier balance and recombination zones
would be major factors limiting the operational stability of OLEDs. However, we think
that the polar EML can cause charge accumulations at both interfaces of the EML. This
is one of the significant differences between the previous works and this work, i.e., the
EML polarizations will be more effective in improving the OLED stability compared to
the SOP of ETLs. The proposed host-molecular design strategy provides the method for
simultaneous control of carrier transport and film polarization to achieve high OLED
performances. Although the impact of SOP on OLED performance has not yet been
isolated, we believe that this manuscript's findings will help to establish a new aspect of
OLED performance from the viewpoint of molecular designs aimed at eliminating SOPs.
We added the additional sentences, the figures, and the tables as follows:

**[Additional sentences]**

- ● [Page 4, line 27] Further, the carrier transport properties of the nonpolar EML were
simultaneously improved due to suitable energy level differences and molecular
packings between host and guest molecules.
- ● [Page 10, line 19] To demonstrate the wide applicability of non-polarization by
codeposition with 1DPCz, another polar TADF molecule, 1,2,3,4-tetrakis(carbazol-
9-yl)-5,6-dicyanobenzene (4CzPN), was also tested. **Supplementary Fig. 12** shows
the thickness dependence of surface potential of a neat 4CzPN and a 10 mol%
4CzPN:host films. The GSP slope values of the neat 4CzPN film and the codeposited
film with 1DPCz were +55 mV nm⁻¹ and +1.6 mV nm⁻¹, respectively
(**Supplementary Table 3**). These results clearly indicate that the 1DPCz's ability to
cancel out SOP is widely applicable as an EML host. In the case of the 2DPCz and

3DPCz hosts, the SOP polarities were same as the films with HDT-1, i.e., the
 4CzPN:2DPCz film and the 4CzPN:3DPCz films showed negative and positive
 SOPs. The J - V characteristics (**Supplementary Fig. 13**) of the 4CzPN-OLED with
 the 2DPCz host also showed higher turn-on voltage derived from the negative SOP
 of the codeposited film (see **Supplementary Note 1**).

**[Additional figure]**

**Supplementary Fig. 12.** Thickness dependence of the surface potential of 4CzPN-based
 vacuum-deposited films. The inset is the molecular structure of 4CzPN.

**[Additional table]**

**Supplementary Table 3.** GSP slope values of the 4CzPN-based films.

	GSP slope (mV nm^{-1})
Neat 4CzPN	+55
4CzPN:1DPCz	+1.6
4CzPN:2DPCz	-35
4CzPN:3DPCz	+16

**Reply to the comments for Reviewer #1**

**Reviewer #1 (Remarks to the Author):**

This work by Tanaka et al. addresses film polarization by orientation of polar molecules
in organic layers and its effect on OLED efficiency and lifetime. Specifically, the authors
have engineered the emissive guest-host system consisting of a polar TADF emitter and
host materials with different polarization. Surprisingly they find that the combination of
the polar emitter with the host having the highest molecular dipole moment and the lowest
glass transition temperature results in the longest-lived OLEDs. This is remarkable and
not expected from the conventional view of OLED stability requiring high-Tg materials.
However, one has to be aware that the observed differences are not tremendously big and
that it is not straightforward to pin them down to a single origin. To be more precise:
OLED efficiency and lifetime are depending on several factors that need to be well
balanced. Here the authors mainly focus on polarization of the emission layer, which will
result in charge accumulation at the interface to the hole (electron) transport layer. This
kind of charge pile-up is one source of efficiency loss, but it is effective mostly at low
voltage or even below turn on of the OLED. What is more relevant in the range where
OLEDs are typically operated, is charge distribution and the resulting recombination
profile due to injected charges in the EML. These two effects are not clearly distinguished
in the paper.

The GSP variations between different hosts are anyway small -- except for 2DPCz. Thus,
it is hard to believe that GSP is the only factor that is responsible for the longer lifetime
of the OLED with 1DPCz as host. It could well be an interplay of several factors, like
energy levels, charge mobility, etc., which leads to a different charge distribution in the
OLED EML.

I think one would need to do a more detailed analysis, including e.g. device simulation
and emission zone analysis to pin down the origin of device improvements.

**[Reply]:**

Firstly, we sincerely thank the reviewer for taking her/his precious time to evaluate our
manuscript. We carefully read your comments and addressed all your concerns according
to your insightful comments and suggestions. Although the reviewer mentioned that the
effect of SOPs in device performance can be observed at only low bias, the recent reports
(Refs. 13 and 15) clarify that a positive SOP of ETLs induces the EQE decrease at the
current density $\sim 0.1 \text{ mA cm}^{-2}$ and low stability due to the SOP-induced exciton quenching.
We note that this event occurs at higher than the turn-on voltage. Furthermore, we agree

that carrier balance and recombination zones can be major factors limiting OLED stability.
However, we think that the polar EML can cause charge accumulations at both interfaces
of the EML. This is one of the significant differences between the previous works and
this work, i.e., the EML polarizations will be more effective in improving the device
stability compared to the SOP of ETLs. The discussion of the SOP effect on OLED
stability is still limited. However, our findings will help to establish a new aspect of OLED
performance from the viewpoint of molecular designs aimed at eliminating SOPs. We put
an additional explanation to clearly state that the effect of SOP in OLED performances
and the carrier balance is also effective in the device stability as follows:

**[Additional sentences]**

[revised manuscript text omitted]

**[Comment #1]:**

- The sentence "However, neither molecular design nor deposition method has been
reported to actively control SOP." in the abstract is not justified. There have been several
reports on the control of SOP, and some of them are cited in the paper as well (e.g. 13, 17,
20 ...).

**[Reply]:**

We thank you for your comment. We revised the sentence in the abstract to state the host
molecular design as follows:

**[Revised sentence]**

However, the report of host molecular design to control SOPs has been limited.

**[Comment #2]:**

- Single carrier devices are not analyzed in much detail. What about SCLC and carrier
mobilities?

**[Reply]:**

We agree that electrical properties such as detailed carrier mobilities are critically
important for discussing fundamental OLED physics. However, since our HOD/EOD
does not form an ohmic contact, it is hard to estimate the accurate carrier mobilities in the
single-layer structures. However, the J-V characteristics reflect the actual injection
barriers between the organic layers, such as HTL/EML or EML/ETL. We believe that
comparing J-V characteristics of the HOD/EODs can be used to discuss OLED
performance.

**[Comment #3]:**

- It doesn't make sense to use different scaling for the vertical axes of IVL characteristics
-- one being linear and the other logarithmic, although this is often done in the literature.
It is impossible to correlate one property (I) with the other (L) in such graphs.

**[Reply]:**

We thank you for your suggestive comment. We put the JVL characteristics
(**Supplementary Fig. 6**) with the vertical axis in the logarithmic scale. This figure clearly
shows that the relatively high turn-on voltage of the 2DPCz:HDT-1 OLED relates to the
suppressed carrier injection, as mentioned in the main text (page 7, line 10).

**[Additional figure]**

**Supplementary Fig. 6.** *J-V-L* characteristics of the OLEDs based on HDT-1.

**[Comment #4]:**

- I don't understand the argument with the different kinetic energy of molecules having
different T_g 's. The kinetic energy of molecules on a surface is given by the substrate
temperature and thus $E_{kin} \sim k_B * T_S$ (or the like) is independent of the T_g . What the
authors actually mean is surface diffusivity, I guess.

**[Reply]:**

We thank you for your insightful comment. We revised the sentences about surface
diffusivity.

**[Revised sentences]**

● [Page 8, line 26] We suppose that the cancellation of the SOP by the 1DPCz matrix
originates from the high molecular diffusivities of 1DPCz molecules, inducing the
random orientation. It has been pointed out that the ratio of substrate temperature (T_s)
and glass transition temperature (T_g), *i.e.*, T_s/T_g , is one of the indicators correlating
to the molecular diffusivity of molecules on a substrate during a vacuum deposition
process. Therefore, we can expect high molecular diffusivities for a low T_g molecule
and *vice versa* at a constant T_s value^{21,27–30}. Previous studies about the orientation of
transition dipole moment (TDM) of a deposited molecule revealed that a high T_g
induces a highly horizontal TDM orientation of a molecule, while a low T_g molecule
randomizes the molecular orientation^{21,28,30}, indicating that excess molecular
diffusivity induces the random orientation of the deposited molecules. The T_g s of
1DPCz, 2DPCz, and mCBP were found to be 60, 140, and 90 °C, respectively
(**Supplementary Table 2**). Note that the T_g of 3DPCz was not clearly observed due
to the high melting point which is close to the decomposition temperature. Thus, the
low T_g of 1DPCz enabled the random PDM orientation in the neat film, leading to
the almost zero GSP (**Fig. 3a**). Similarly, the 1DPCz molecules form the nearly-
random orientation in a codeposited film with HDT-1. Since 1DPCz possesses high
diffusivities due to the small molecular size, the 1DPCz molecules effectively cancel
out the polarization of the HDT-1 (**Fig. 3c**). Further, 2DPCz is also a polar molecule
with a comparable PDM magnitude to 1DPCz. However, the high T_g of 2DPCz, *i.e.*,
low molecular mobility, inhibits the canceling of the SOP in the HDT-1:2DPCz film.

**[Comment #5]:**

- Correlation between PDM and TDM alignment, where the latter is taken from angular
PL measurements, is not a straightforward task. One needs to know the relation between

them on the molecules, as obtained e.g. by DFT calculations. The authors have only
shown the PDM calculations in their manuscript.

**[Reply]:**

We thank you for your comment. We apologize for the incomplete information. According
to the calculation result, the TDM orientation of HDT-1 can be estimated that the vector
is oriented parallel to the central phenyl ring. We put the additional sentence and the figure
as follows:

**[Additional sentence]**

● [Page 9, line 25] Note that we estimate that the TDM direction of HDT-1 is oriented
parallel to the central phenyl ring (**Supplementary Fig. 9b**).

**[Additional figure]**

**Supplementary Fig. 9.** Orientation of transition dipole moment of HDT-1. **a** Angular
dependence of the PL intensity of deposited films of HDT-1:mCBP and HDT-1:1DPCz.
**b** Calculated direction of transition dipole moment for HDT-1.

**[Comment #6]:**

- Differences in EQE between the mCBP and 1DPCz hosts are rather small. I am
wondering how large the error bars of the single measurements are and how the statistics
of several nominally identical OLED devices is.

**[Reply]:**

The OLEDs were fabricated in the same batch using the method to replace several shadow
masks, meaning that the organic stacks, except for the EMLs, are identical. This is one of
the reasons to compare such small differences directly. In addition, we checked the batch-
to-batch reproducibility of device EQEs. The standard deviation of EQE was less than

0.3%. We think that the small deviation in our device fabrication and measurement
systems can clarify the small differences in the performance of the OLEDs. Furthermore,
the improvement of the device stability was also confirmed in the case of the initial
luminance of $1,000 \text{ cd m}^{-2}$. We put the additional sentences and the supplementary figures
as follows:

**[Additional sentences]**

- ● [Page 11, line 12] We note that these devices were fabricated in the same batch using
the method of replacing several shadow masks for the EMLs, meaning that the
organic layers, except for the EMLs in each device, are identical. Further, the
relatively small batch-to-batch reproducibility of EQE was also confirmed
(**Supplementary Fig. 14**). Thus, we can identify the small EQE difference as a
significant issue.
- ● [Page 13, line 14] The trend of the device stability was confirmed in the case of the
measurement with the same initial luminance (L_0) of $1,000 \text{ cd m}^{-2}$ (**Supplementary**
**Fig. 18**).

**[Additional figures]**

**Supplementary Fig. 14.** Current density-EQE profiles of the same OLEDs in different
batches. The device structure was ITO/HAT-CN (10 nm)/NPD (40 nm)/TCTA (10
255 nm)/1DPCz (5 nm)/1DPCz:mCBP:HDT-1:v-DABNA (40 nm)/SF3-TRZ:Liq (30
256 nm)/Liq (2 nm)/Al. The standard deviation of EQE values was less than 0.3% in the whole
range of current density.

**Supplementary Fig. 18.** Device stability of the HDT-1-based OLEDs. Luminance decay
profiles measured under constant current density (initial luminance: $1,000 \text{ cd m}^{-2}$).

**Reply to the comments for Reviewer #2**

**Reviewer #2 (Remarks to the Author):**

The authors report a strategy to achieve high operational stability of OLEDs by reducing
spontaneous orientation polarization (SOP) and improving carrier balance in the emission
layer (EML). They found 1DPCz, which has a high molecular kinetic energy and
moderate permanent dipole moment (PDM), efficiently eliminates the total SOP in the
EML incorporating a polar TADF emitter. The hole conductance of this material is higher
than that of the reference material, mCBP, while the electron conductance is comparable.
The operational stability as well as the maximum EQE of the OLED are improved by
using 1DPCz as a host material, even though T_g of this material is relatively low. They
further applied this molecule to the host of a blue TAF-OLED, and demonstrated its high
operational stability. The results presented in this manuscript are interesting and would
benefit the OLED community; however, the underlying mechanisms are still ambiguous
(see comments below). Moreover, a similar work has been published recently, so that the
impacts of this work would be moderate. Overall, the reviewer considers that this work
does not meet the high standard of Nature Communications.

**[Reply]:**

Firstly, we sincerely thank the reviewer for taking her/his precious time to evaluate our
manuscript. We carefully read your comments and addressed all your concerns according
to your insightful comments and suggestions. A similar work reported by E. Afolayan et
al. (Ref. 15) clarifies that a positive SOP of ETLs can be reduced by codeposition with
MDPE, and the reduced SOP improves the device's stability. The major differences
between the reported work and this work are that we studied an EML polarization, and
the EML polarization was canceled by the deposition with a designed host molecule. This
host molecule maintains the emission performances of an emitter molecule, indicating
that this method can be applied to EML designs successfully. Furthermore, we think that
the polar EML can cause charge accumulations at both interfaces of the EML, meaning
that the polarization control of EMLs is highly effective in improving the device
performance compared to ETL polarizations. Although we agree that the carrier
balance/recombination zone would be a major factor limiting the device stability, our
study provides a universal strategy to design EML polarizations using conventional
carbazole-based hosts actively. We believe that our findings will help to establish a new

aspect of OLED performance from the viewpoint of molecular designs aimed at
minimizing SOPs.

**[Comment #1]:**

1. A similar work has been recently published (E. Afolayan et al., Phys. Rev. Applied,
10.1103/PhysRevApplied.17.L051002). The authors should refer to this paper and state
the significance and novelty of this work.

**[Reply]:**

We appreciate your suggestive comments. We referred to the paper as Ref. 15. The paper
reported by E. Afolayan et al. demonstrated that reducing polarization in an electron
transport layer (ETL) dramatically improves OLED performance, such as EQE and
device stability. They used a dilution method with MDPE for reducing SOP in a TPBi
ETL. The dilution of the ETL suppresses the exciton-polaron quenching induced by
accumulated interface charges due to the SOP. Thus, the reported method is beneficial to
control the SOP in carrier transport layers such as an HTL and an ETL. However, the
dilution might be difficult to apply an EML because precise control of emission
performance of emitter molecules such as TADF is necessary to maintain ultimately high
OLED performance. The impact of the MDPE dilution of EMLs on emission properties
such as PLQYs and exciton lifetimes is still unknown. In this work, we demonstrated
nonpolar EML using host molecular designs. The molecules are based on a carbazole unit,
which is well-used as conventional host molecules, such as CBP and mCBP. We found
that the designed molecule, 1DP Cz, can clearly reduce the SOP of the deposited film with
polar TADF molecules, and maintain the emitter's PLQY and the exciton lifetime. The
findings will allow emitter molecules to exhibit their underlying potential, improving
OLED efficiency and stability. We put the additional sentences to mention the difference
between the Afolayan's work and this work as follows:

**[Additional sentences]**

● [Page 4, line 9] Another proposed method to cancel out the SOP of an electron
transport layer was diluting polar molecules with medium-density polyethylene
(MDPE). The MDPE-codeposition reduced the SOP of a TPBi-ETL and improved
the OLED stability due to the suppressed EPQ in a fluorescent EML¹⁵. This method
would be beneficial to cancel out the SOPs of organic layers. However, it may be
difficult to apply to EMLs because the EML designs in conventional OLEDs are

based on the precise controls of exciton and carrier performance such as
fluorescence/phosphorescence yields, exciton confinement, carrier recombination,
and carrier transport.

**[Comment #2]:**

2. (Fig. 4 and Fig. 5, p.10 and p.11) The authors discussed the origin of improvement of
the device performance; however, the influence of reducing SOP on the device
performance is still a speculation. The authors should estimate the contributions of the
carrier balance (or recombination zone) and SOP to the improvements of EQE and
lifetime, quantitatively. For instance, although the GSP slope of the 1DPCz:HDT-1 film
is very small (1.5 mV nm^{-1}), that of the mCBP:HDT-1 film is also small (5.2 mV nm^{-1}).
The reviewer is wondering whether this difference (corresponding to $\sim 0.1 \text{ mC m}^{-2}$) can
lead to the higher EQE (Fig. 4a) and the significant improvement of the operational
stability (Fig. 4b, Fig. 5d).

**[Reply]:**

We thank you for your insightful comment. We agree that the SOP difference between the
EMLs with different hosts was relatively small, and the carrier balance also strongly
affects the device stability. We think that although the impact of SOPs on device stability
is minor compared to the carrier balance in our OLEDs, SOP also has no small effect on
device stability. Several studies, such as E. Afolayan's work, reported that ETLs with a
positive GSP induce hole accumulations at the EML interface; in contrast, an EML
polarization can cause the charge accumulations at both interfaces of the EML, resulting
in severe exciton annihilations, leading to a decrease of device performances. We
additionally tested 4CzPN-based OLEDs with 1DPCz/2DPCz/3DPCz hosts with
relatively large SOP differences of the codeposited EML. We barely observed a small
amount of the exciton quenching by SOP-induced accumulated charges. Although the
impact of the SOP on initial device performance is rather small, the SOP-induced
phenomena have no small effect on long-term performance such as device stability. We
put the additional sentences and the figures about the results of 4CzPN devices as follows:

[Additional sentences]

- ● [Page 10, line 19] To demonstrate the wide applicability of non-polarization by
codeposition with 1DPCz, another polar TADF molecule, 1,2,3,4-tetrakis(carbazol-
9-yl)-5,6-dicyanobenzene (4CzPN), was also tested. **Supplementary Fig. 12** shows
the thickness dependence of surface potential of a neat 4CzPN and a 10 mol%
4CzPN:host films. The GSP slope values of the neat 4CzPN film and the codeposited
film with 1DPCz were $+55 \text{ mV nm}^{-1}$ and $+1.6 \text{ mV nm}^{-1}$, respectively
(**Supplementary Table 3**). These results clearly indicate that the 1DPCz's ability to
cancel out a SOP is widely applicable as an EML host. In the case of the 2DPCz and
3DPCz hosts, the SOP polarities were same as the films with HDT-1, i.e., the
4CzPN:2DPCz film and the 4CzPN:3DPCz films showed negative and positive
SOPs. The *J-V* characteristics (**Supplementary Fig. 13**) of the 4CzPN-OLED with
the 2DPCz host also showed higher turn-on voltage derived from the negative SOP
of the codeposited film (see **Supplementary Note 1**).
- ● [Page 12, line 23] The EQE performances of OLEDs based on the 4CzPN:host EMLs
with larger differences in the SOPs are shown in **Supplementary Fig. 16a**. The
4CzPN-based OLEDs with different hosts exhibited comparable EQE values because
of the similar PLQY (34-38%) of the codeposited films. The effects of the EPQ
caused by the SOP-induced accumulated charges were investigated by a DCM and a
bias-dependent PL measurements (**Supplementary Fig. 16b-f** and **Supplementary**
**Note 2**). We confirmed a quite small decrease in the PL intensity under the voltage
range from the carrier accumulation voltage to the actual current injection voltage,
and estimated that this PL decrease originates from the exciton quenching by the
accumulated holes at the EML interface. The reason for the quite small change in the
PL intensity is not only the small amount of the quenched excitons but also the
thicker EML than that of the previous studies, resulting in a decrease in the
percentage of the photogenerated excitons quenched at the limited quenching site
near the interface (**Supplementary Fig. 17**). Furthermore, because a relatively long
exciton lifetime (μs - ms) of TADF emitters is strongly influenced by a voltage applied
across the film³⁵, a shape of the bias-dependent PL profiles contains a portion of
exciton quenching via an applied electric field (**Supplementary Fig. 16c-e**)³⁶. Thus,
it might be necessary to apply some techniques for the precise identification of the

percentage of SOP-induced exciton quenching in TADF-based OLEDs by the bias-
dependent PL measurement.

● [Page 13, line 27] Although we could not clearly observe the interfacial charge
accumulations (especially for the 2DPCz-based EMLs with the negative
polarizations) and the exciton quenching through EPQ by the DCM and the bias-
dependent PL measurement, a little effect of the SOP results in the negative impact
on the device stability due to continuous long-time device operations. Another factor
affecting the device stability is the carrier transport properties of the EMLs. The
balanced carrier transports suppress EPQ, leading longer device lifetimes. As
discussed above, the HDT-1:1DPCz-EML possesses a better carrier balance due to
the high hole transport and the moderate electron transport properties, reducing the
charge density near the EML interfaces.

[Additional figures]

**Supplementary Fig. 13.** 4CzPN-based OLEDs. a OLED structure. b Voltage-current
density-luminance characteristics of the 4CzPN-based devices. c HOMO-LUMO
diagrams of 4CzPN and the host molecules.

**Supplementary Fig. 16.** OLED performances of the 4CzPN-based OLEDs. **a** EQE-
 current density profiles. **b** DCM profiles. **c-e** Bias-dependent photoluminescence
 intensity of the OLEDs with 1DPCz (**c**), 2DPCz (**d**), and 3DPCz (**e**). **f** Photocurrent
 characteristics.

**Supplementary Fig. 17.** Effect of EML thickness in the bias-dependent
 photoluminescence measurement. **a** The case of a thin EML. **b** The case of a thick EML.
 **c** Absorption spectra of the films.

 **Supplementary Fig. 19.** Device stability of the 4CzPN-based OLEDs. Luminescence decay
 profiles measured under constant current density (initial luminance: 1,000 cd m⁻²).

**[Comment #3]:**

3. The SOP of ETL often facilitates the charge accumulation at the EML/ETL interface.
 The authors should also examine the GSP of SF3-TRZ and SF3-TRZ:Liq films.

**[Reply]:**

We thank you for your comment. Because SF3-TRZ is almost nonpolar, the GSPs of the
 films are small. We put the additional sentence and the figure showing the results of GSP
 measurements as follows:

**[Additional sentence]**

● [Page 10, line 15] We note that since the SOPs of the SF3-TRZ and the SF3-TRZ:Liq
 codeposited films are small compared to the EMLs in this study (**Supplementary**
 **Fig. 11**)²⁷, the SOPs of the EMLs mostly affect the carrier injection properties of the
 OLEDs.

**[Additional figure]**

 **Supplementary Fig. 11.** Thickness dependence of the surface potentials of SF3-TRZ and
 SF3-TRZ:Liq films.

**[Comment #4]:**

4. The title of the paper should be reconsidered. Although the enhancement of the lifetime
of a blue OLED is demonstrated, the results and discussions about the "blue OLED" are
rather additional (as the authors indicate in the introduction). Further, the contribution of
"film polarization" to the operational stability is still unclear (as suggested in the second
comment).

**[Reply]:**

We thank you for your comment. We revised the title as follows:

**[Revised title]**

**Simultaneous control of carrier transport and film polarization of emission layers aimed**
**at high performance of organic light-emitting diodes**

**[Comment #5]:**

5. The improvement of the carrier balance in EML is also suggested as an important factor,
in terms of the host selection strategy. The authors should discuss the origin of the
enhanced hole conductivity in the HDT-1:1DPCz(2DPCz) film as well as the reduced
electron conductivity in the HDT-1:2DPCz(3DPCz).

**[Reply]:**

We thank you for your insightful comment. We estimate that the difference in hole
conductivity originated from the HOMO energy gap between the host and the guest
molecules. The slightly larger HOMO gaps limit the hole transport in the mCBP- and
3DPCz-based layers. In contrast, the electron transport in these layers is limited by
intermolecular LUMO overlap because the LUMO gap between the host and guest
molecules is significantly large. The LUMO distribution of the molecules is relatively
localized at the center of the molecule. Therefore, the electron transport of the molecules
possessing bulky units such as 2DPCz and 3DPCz is suppressed. We put the additional
sentences and the figures showing calculated HOMO/LUMO distributions as follows:

**[Additional sentences]**

● [Page 6, line 11] **Since the hole transport of the neat HDT-1 film was low compared**
**to the deposited films with 1DPCz and 2DPCz, the better hole transport properties**

can be attributed to the HOMO level differences between the host and the emitter
molecules. The HOMO levels of 1DPCz and 2DPCz are shallower compared to that
of HDT-1 (**Supplementary Fig. 3**), indicating that the hole transport can be governed
by the hole transport of 1DPCz and 2DPCz with the shallower HOMO level, leading
to the improved hole current. In contrast, the HOMO levels of mCBP and 3DPCz
were comparable to that of HDT-1, thus, the J - V curves of the HODs with these host
molecules and the neat TADF were also comparable. On the other hand, the electron
transports of the host:guest systems were considerably suppressed compared to the
EOD based on the neat HDT-1 film because of the HDT-1's deep LUMO level to
limit the electron transport due to the electron traps in the codeposited films. The
reason for the variation in the electron transport properties of the codeposited films
would be originating to molecular packing. The calculated HOMO and LUMO
distributions of the host and the guest molecules are shown in **Supplementary Fig.**
**4**. Although the HOMO is widely distributed through the host molecules because of
the donor Cz moieties, the LUMO distributions are relatively located on the center
of the molecules. Thus, because large intermolecular LUMO overlaps are necessary
for efficient intermolecular electron transfer, sufficient molecular packing is
beneficial to enhance the electron transport in amorphous films²⁶. Therefore, host
molecules with large molecular sizes, such as 2DPCz and 3DPCz with the bulky
donor Czs, have a slight disadvantage for efficient electron transport in the
codeposited films.

[Additional figure]

Supplementary Fig. 4.
Calculated HOMO/LUMO
distributions. a HDT-1, b
mCBP (conformer #1), c
mCBP (conformer #2), d
1DPCz, e 2DPCz, f 3DPCz.
Note that mCBP possesses
stable conformers, such as
conformers #1 and #2.

[Comment #6]:

6. Ref. 11 and 32, 19 and 27 are the same.

[Comment #7]:

7. (Figure 1 caption) HDT-a should be HDT-1.

[Reply]:

We apologize for our carelessness. We rebuilt the references correctly, and revised the
caption correctly.

REVIEWER COMMENTS

Reviewer #1 (Remarks to the Author):

The authors have done a careful and quite extensive revision of their manuscript to address all of the reviewers' comments and concerns. And, even if some of the raised issues could not be fully clarified with absolute certainty, I am satisfied with the added text and discussion in the revised manuscript as well as the additional information provided.

This manuscript is definitely an important first step for investigating the effect of SOP in the emission layer of an OLED and will certainly trigger further studies in that direction.

For that reason I recommend acceptance of the revised manuscript.

Reviewer #2 (Remarks to the Author):

The authors have extensively revised the manuscript, resulting in a significant improvement in the overall quality compared to the previous version. In particular, they demonstrate the validity of their strategy in OLEDs using another emitter, 4CzPN, which exhibits significant SOP. Their host material, 1DPCz, effectively eliminates the SOP of 4CzPN and extends the device lifetime. Additionally, they clarify the difference between this work and the previous work by Afolayan et al., highlighting that this study focuses on the SOP of the emission layer, which can contribute to device lifetime even if it is small. Although this work presents a new aspect of the design and selection strategy of host materials for improving device performance, I am still not convinced that even small SOP of EML indeed affects the device lifetime. I encourage the authors to estimate the contribution of SOP to device performance, for example, by modeling the degradation process with and without SOP in the EML.

I also suggest a few minor points:

1. In the abstract, the authors mention "leading to high operational stability compared to ... CBP and mCBP". However, in this manuscript, only mCBP has been tested in the device form. It is more accurate to remove "CBP".

2. It would be helpful for readers if the energy level diagrams (Supplementary Fig. 3) were included in Fig. 2 for easier reference and understanding.

3. On page 11, "the relatively small batch-to-batch reproducibility of EQE" could be changed to "the relatively small batch-to-batch variation of EQE".

Reply to the comments

First, we sincerely thank the reviewers for taking their precious time to evaluate our manuscript and providing insightful comments and suggestions. Based on their comments, we have revised the manuscript.

Reply to the comments for Reviewer #1

Reviewer #1 (Remarks to the Author):

The authors have done a careful and quite extensive revision of their manuscript to address all of the reviewers' comments and concerns. And, even if some of the raised issues could not be fully clarified with absolute certainty, I am satisfied with the added text and discussion in the revised manuscript as well as the additional information provided.

This manuscript is definitely an important first step for investigating the effect of SOP in the emission layer of an OLED and will certainly trigger further studies in that direction.

For that reason, I recommend acceptance of the revised manuscript.

[Reply]:

We sincerely thank the reviewer for the positive assessment.

Reply to the comments for Reviewer #2

Reviewer #2 (Remarks to the Author):

The authors have extensively revised the manuscript, resulting in a significant improvement in the overall quality compared to the previous version. In particular, they demonstrate the validity of their strategy in OLEDs using another emitter, 4CzPN, which exhibits significant SOP. Their host material, 1DPCz, effectively eliminates the SOP of 4CzPN and extends the device lifetime. Additionally, they clarify the difference between this work and the previous work by Afolayan et al., highlighting that this study focuses on the SOP of the emission layer, which can contribute to device lifetime even if it is small. Although this work present a new aspect of the design and selection strategy of host materials for improving device performance, I am still not convinced that even small SOP of EML indeed affects the device lifetime. I encourage the authors to estimate the contribution of SOP to device performance, for example, by modeling the degradation process with and without SOP in the EML.

[Reply]:

We have carefully read your comments and addressed your concerns based on your insightful comments and suggestions. We considered the impact of charge accumulation on device degradation by using a theoretical model. Although the model could not cover the entire phenomena in a degraded OLED, we confirmed that a small SOP results in the improvement of device stability. To investigate the effect in more detail, it is necessary to consider the effect of all defect formation processes and changes in carrier balance and recombination zone during device operation in the future. We added the sentences and the figure as follows:

[Additional sentences]

Supplementary Note 3: Demonstration of impact of charge accumulation

To demonstrate the impact of exciton-polaron annihilation on device stability, we simulated device degradation using rate equations². The singlet (S) and triplet (T) exciton densities of TADF-OLEDs under electrical excitation can be expressed as,

$$\frac{dN_S}{dt} = -(k_r + k_{ISC})N_S + k_{RISC}N_T - k_{SP}N_SN_P - k_{SQ}N_SN_Q + \frac{0.25J}{de} \left(1 - \frac{N_Q}{N_{mol}}\right), \quad (1)$$

$$\frac{dN_T}{dt} = k_{ISC}N_S - k_{RISC}N_T - k_{TP}N_TN_P - k_{TQ}N_TN_Q + \frac{0.75J}{de} \left(1 - \frac{N_Q}{N_{mol}}\right), \quad (2)$$

where N_S , N_T , N_P , and N_Q represent the densities of singlet excitons, triplet excitons, polarons (P), and degradation defects (Q), respectively. k_r , k_{ISC} , and k_{RISC} denote the radiative decay rate from singlet state, the intersystem crossing (ISC) rate from singlet to triplet states, and the reverse ISC (RISC) rate from triplet to singlet states, respectively. k_{SP} , k_{SQ} , k_{TP} , and k_{TQ} are the rate coefficients of S-P, S-Q, T-P, and T-Q annihilations, respectively. J , d , e , and N_{mol} are the current density, recombination zone width, electron charge, and molecular density of the emission layer, respectively. For simplicity, the terms of singlet-triplet, triplet-triplet annihilations, and nonradiative decay from S and T states were ignored in Eqs. (1) and (2). We assume that degradation defects are generated only via exciton-polaron annihilations such as SPA and TPA, which induce exciton quenching (S-Q and T-Q quenching) and nonradiative recombination to reduce the EL intensity of degraded OLEDs. Thus, the change in N_Q is directly related to the decrease in EL intensity during the OLED operational test. The defect formation rate ($dN_Q/d\tau$) to operational time (τ) is assumed to be given by

$$\frac{dN_Q}{d\tau} = \alpha_S k_{SP} N_S N_P + \alpha_T k_{TP} N_T N_P, \quad (3)$$

where α_S and α_T denote the probability factors of defect formation via SPA and TPA, respectively. For steady-state operation ($dN_S/dt = dN_T/dt = 0$), N_S and N_T can be written as:

$$N_S = \frac{J}{de} \left(1 - \frac{N_Q}{N_{mol}}\right) \frac{0.25(k_{TP}N_P + k_{TQ}N_Q) + k_{RISC}}{(k_{ISC} + k_r + k_{SP}N_P + k_{SQ}N_Q)(k_{RISC} + k_{TP}N_P + k_{TQ}N_Q) - k_{RISC}k_{ISC}}, \quad (4)$$

$$N_T = \frac{1}{k_{RISC} + k_{TP}N_P + k_{TQ}N_Q} \left(k_{ISC}N_S + \frac{0.75J}{de} \left(1 - \frac{N_Q}{N_{mol}}\right) \right). \quad (5)$$

For simplicity, we assumed that the carrier balance is unity, indicating that polarons interacting with excitons originate from SOP-induced accumulated charges at the EML interfaces. Thus, using Eqs. (3)–(5), the N_P dependence of the defect formation rate per unit τ at the initial stage of operational degradation ($N_Q = 0$ at $\tau = 0$) can be depicted in **Supplementary Fig. 19a**, indicating that the increase in N_P and the decrease in k_{RISC} cause severe defect formation via exciton-polaron annihilations. Finally, we depicted N_P

dependence of normalized change in N_S after device operation for unit time τ ($N_S / N_S(\tau = 0)$) in **Supplementary Fig. 19b**, assuming negligible k_{SQ} and k_{TQ} . In our previous research, we confirmed a small change in the PL intensities of pristine and degraded TADF-OLEDs compared to those in the EL intensity³, indicating that the degradation defects act as nonradiative recombination sites rather than exciton quenchers (k_{SQ} and $k_{TQ} \sim 0$). Since we assumed that the exciton annihilations with SOP-induced accumulation polarons are the only sources of defect formation, $N_S / N_S(\tau = 0)$ values were normalized by the values in the case of a quite small charge density, $N_P = 0.016 \text{ mC m}^{-2}$. **Supplementary Fig. 19b** indicates that a small accumulation charge density and a large k_{RISC} improve the operational lifetime. We note that this demonstration ignored detailed conditions such as changes in carrier balance and recombination zone during the degradation due to the generated defects (carrier traps), while we have revealed that these factors also impact the reduction in EL intensity in degraded TADF-OLEDs⁴. Therefore, the N_P dependence of device stability would become clearer using a more precise simulation method.

[Additional Figure]

Supplementary Fig. 19. Demonstration of impact of SOP-induced charge accumulation on OLED degradation. **a** Accumulated charge density dependence of relative defect formation rate per unit operational time. **b** Accumulated charge density dependence of normalized change in singlet exciton density ($N_S / N_S(\tau = 0)$) during device operation. The $N_S / N_S(\tau = 0)$ values were normalized by the value with a small charge density of $N_P = 0.016 \text{ mC m}^{-2}$. The k_{RISC} values of the TADF emitters were assumed to be approximately $9.2 \times 10^5 \text{ s}^{-1}$ for HDT-1 and $1.0 \times 10^5 \text{ s}^{-1}$ for 4CzPN. To depict these figures, the values of J , d , N_{mol} , k_r , k_{ISC} , k_{SP} , and k_{TP} were assumed to be 3.3 mA cm^{-2} , 5

nm, $6 \times 10^{20} \text{ cm}^{-3}$, $3.7 \times 10^7 \text{ s}^{-1}$, $8.7 \times 10^7 \text{ s}^{-1}$, $1.0 \times 10^{-11} \text{ cm}^3 \text{ s}^{-1}$, and $1.0 \times 10^{-11} \text{ cm}^3 \text{ s}^{-1}$, respectively.

[Comment #1]:

1. In the abstract, the authors mention "leading to high operational stability compared to ... CBP and mCBP". However, in this manuscript, only mCBP has been tested in the device form. It is more accurate to remove "CBP".

[Reply]:

We appreciate your comments. We revised the sentence as follows:

[Revised sentence]

- [Page 2, line 16] We found that a small-sized polar host molecule that possesses both high molecular diffusivities and moderate permanent dipole moment, such as 3,6-diphenylcarzole-9-ylbenzene (1DPCz), well cancels out the polarization formed by the SOP of the TADF molecules in the EML without a disturbance of the TADF molecules' intrinsic orientation trend, leading to high operational stability compared to those with the conventional carbazole-based host molecules such as 3,3'-di(9H-carbazol-9-yl)-1,1'-biphenyl (mCBP).

[Comment #2]:

2. It would be helpful for readers if the energy level diagrams (Supplementary Fig. 3) were included in Fig. 2 for easier reference and understanding.

[Reply]:

We thank you for your comment. We revised Fig. 2 as follows:

[Revised Figure]

Fig. 2. Carrier transport properties of codeposited films. **a** Current density-voltage characteristics of hole-only devices. **b** Current density-voltage characteristics of electron-only devices. **c** HOMO-LUMO energy diagrams. **d** Device structure of OLEDs. **e** Current density-luminance-voltage characteristics of OLEDs.

[Comment #3]:

3. On page 11, "the relatively small batch-to-batch reproducibility of EQE" could be changed to "the relatively small batch-to-batch variation of EQE".

[Reply]:

Thank you for pointing it out. We revised the sentence as follows:

[Revised sentence]

- [Page 10, line 3] Further, the relatively small batch-to-batch variation of EQE was also confirmed (Supplementary Fig. 13).

REVIEWERS' COMMENTS

Reviewer #2 (Remarks to the Author):

The authors have addressed all the issues raised by the reviewers. I recommend the publication of this paper in Nature Communications.